# The Role of Supervision in Preventing Burnout among Professionals Working with People in Difficulty

**DOI:** 10.3390/ijerph19010160

**Published:** 2021-12-24

**Authors:** Iasmina Iosim, Patricia Runcan, Virgil Dan, Bogdan Nadolu, Remus Runcan, Magdalena Petrescu

**Affiliations:** 1Economics and Finance Company Department, Faculty of Management and Rural Tourism, Banat’s University of Agricultural Science and Veterinary Medicine “King Mihai I of Romania”, 300645 Timisoara, Romania; iasminaiosim@usab-tm.ro; 2Department of Social Work, The Faculty of Sociology and Psychology, West University of Timisoara, 300223 Timisoara, Romania; patricia.runcan@e-uvt.ro (P.R.); virgil.dan68@e-uvt.ro (V.D.); 3Department of Sociology, The Faculty of Sociology and Psychology, West University of Timisoara, 300223 Timisoara, Romania; 4Department of Pedagogy, Psychology and Social Work, Faculty of Educational Science, Psychology and Social Work, Aurel Vlaicu University of Arad, 310032 Arad, Romania; remus.runcan@uav.ro; 5Department for Teaching Training, West University of Timisoara, 300223 Timisoara, Romania; magdalena.petrescu@e-uvt.ro

**Keywords:** supervision, social workers, clerics, people in difficulty, burnout

## Abstract

The value of people in their various dimensions is a priority in the postmodern era. In this respect, programs are being implemented for disadvantaged social categories to compensate for differences, reduce discrepancies, and integrate marginalized people into society. This, however, is not easy, and the work of professionals with people in difficulty is frequently difficult, consuming multiple resources and, sometimes, leading to burnout. The professions involved in the recovery work of people in difficulty provide social, medical, psychological, and spiritual assistance services in order to restore or increase the well-being of disadvantaged people or social groups. This study presents an analysis of burnout among social workers and clerics and the effect of supervision on burnout. In support of this, a sociological survey (*n* = 502) was conducted on a convenience sample of Romanian social workers and clerics in June 2018. The main conclusion of the study is that supervising professionals working with people in difficulty significantly reduces the risk of burnout.

## 1. Introduction

Supervision can be defined as the act or function of supervising (overseeing a process, work, workers, etc., during the execution of work or performance).

Speaking of the plethora of definitions of supervision in social work, Reference [1] noted: “what is common in these definitions is that they describe social work supervision as a process, activity, and relationship(s), based in an organizational professional and personal mandate, with designated roles, and boundaries, in which particular functions are performed with the aim of facilitating the best/competent service/practice with clients”.

Over time, supervision has seen its meaning change: first, it was a supportive and reflective space for social workers, then it was assimilated with counselling-/psychotherapy-bound models of supervision, before finally moving its focus “from the person doing the work to the work itself” [2].

Its primary functions are “administrative case management, reflecting on and learning from practice, personal support, mediation (in which the supervisor acts as a bridge between the individual staff member and the organization), and professional development” [3].

O’Donoghue emphasized the uniqueness of supervision in social work through the fact that, operating within the paradigm of social work, “social work and social workers view the world and construct the social work language, principles, beliefs, assumptions and methods” [4]. This perspective adds to the dedication to the principles of anti-oppressive and anti-discriminatory practice, of human rights and social justice, and of power and empowerment, giving the social assistance profession a leading place between other professions and disciplines [5,6].

In particular situations, such as the COVID-19 pandemic, the role of supervisors is of particular importance, given that the impacts of the pandemic have been minimized by such interventions as “feedback of information provided by middle managers and supervisors”, organizational culture facilitating interdisciplinary teamwork, and participative leadership styles [7].

In their review of the research on supervision in social work, Reference [1] came to the conclusion that research in this field should “focus on evaluating the effectiveness of supervision practices across all of the various formats in order to develop empirically supported supervision practice models; discover how supervision practice contributes to client outcomes and involve clients in supervision research; develop an international understanding of the nature and practice of supervision; be more widely known within the social work profession”.

In Romania, after 1989, supervision was either not known about in most state institutions or was perceived as control and supervision, mainly because of the mentality of the employees.

Pastoral supervision differs from supervision in social work only from the perspective of the religious/spiritual approach. It is, according to The Association for Pastoral Supervision and Education [8], a boundaried, intentional, planned, and regular space where the supervisor meets the supervisee(s), and a way of growing in accountability, mutual learning, quality of presence, response to challenges, role competence, self-awareness, spiritual/theological reflection, and vocational identity, as well as being attentive to issues of fitness to practice, impact of the work upon all concerned parties, management of boundaries, professional identity, and skill development, based on practice that is, psychologically, and spiritually/theologically informed, and is contextually sensitive.

In Romania, pastoral supervision was either of secondary importance or completely absent among the over 22,000 professionals (chaplains, confessors, faith community nurses or parish nurses, imams, parish workers, pastoral careers, pastoral counsellors, pastors, priests, or rabbis) working in 2015 [9].

Occupational stress [10], compassion fatigue [11], secondary traumatic stress [12], and vicarious trauma [13] can all lead to a syndrome called burnout [14], job burnout [15] or professional burnout [16]—defined as “physical or mental collapse caused by overwork or stress” (Lexico). “Burnout was found to mediate the association of family-to-work conflict] with workplace injuries” [17].

Burnout varies depending on context. During the COVID-19 pandemic, for instance, burnout components in health workers have been predicted by a wide variety of factors, such as: *emotional exhaustion*, by anxiety, the burden of treating suspected COVID-19 patients, depression, fear of infection, marital status, sex; *depersonalization*, by anxiety, endless months of work in the current department, depression, job category, dissatisfaction with work environment; and *personal accomplishment*, by job stress, socioeconomic status, workload of directly interacting with patients [18,19,20,21].

Burnout can also occur in educational environments, particularly during such events as the first wave of the COVID-19 pandemic, for instance [22], when “an increase in the level of educational burnout, a decrease in life satisfaction, and the use of negative strategies of coping with stress were accompanied by a deteriorated mental condition of students, with female respondents scoring higher on the scale of disorders in comparison to males”.

Job demands and resources affect burnout in a different way: job demands are “positively associated with burnout”, while job resources are “negatively associated with burnout”, and relations are “partially mediated by state mindfulness” (“a state of consciousness during which an individual actively engages in purposeful awareness and attention to the present moment” [23])—job demand has a slight negative direct effect on mindfulness, while job resources have a strong, positive direct effect on mindfulness.

Burnout chases away over 50% of U.S.A. clerics in their first five years of work [24], kindergarten teachers leading them to new careers in Spain [25], and almost 50% of Romania’s trained clerics who choose to reconvert professionally or not to work at all.

Burnout caused by prolonged professional stress can be prevented by helping professionals recognize, comprehend and manage work-related stress with professional supervision as additional support [26].

In a comprehensive approach, Sălășan and Rață define the people and groups in difficulty as: refugees seeking protection; asylum applicants expecting to be granted habitation rights willing to actively integrate into the society and customs, including employment; infant carefulness and watchfulness for incapacitated children aged 0–4 and extra-school watch and care kids until 12 years of age; people aged over 55 presenting specific symptoms or being affected by dementia; seniors over 55 years of age requiring assistance in their daily lives; former and currently addicted persons isolated form the active part of the society; former and currently convicted persons having difficulties reintegrating into the society after serving time for unlawful conduct; immigrants; people in long-term unemployment with no defined perspective of re-employment; persons affected by autism, children or adults presenting specific autistic syndrome symptoms combined or not with other forms of incapacities; persons affected by burnout or presenting the distress of burnout symptoms originating from a professional or private environment and where full employment is provisionally not possible; persons presenting incapacities of psychical or physical nature, with challenges or impairments preventing them from complete non-discriminatory interaction with the society; persons previously affected by accidents or diseases resulting in severe brain damage and facing constraints when attempting to act normally in society; youngsters originating from or integrated in education systems dealing with special needs related to psychological challenges and/or interaction issues; youngsters somewhat mentally challenged and requiring assistance and support to actively integrate into society [27].

The professions involved in the recovery of people in difficulty are collected under the generic title of “caring profession” (“a job that involves looking after other people, such as nursing, teaching, or social work”—Lexico) or “helping professions” (“professions that nurture the growth of or address the problems of a person’s physical, psychological, intellectual, emotional or spiritual well-being”) such as education, life coaching, medicine, ministry (through pastoral intervention, spiritually-oriented intervention [28]), nursing, psychological counselling, psychotherapy, and social work (through counselling).

According to Stevenson, in the context of professional supervision, the purpose of this action may lead to: (i) raising awareness of the roles and responsibilities of social workers; (ii) encouraging social workers to pursue professional goals; (iii) increasing the capacity to understand people, problems, and situations; (iv) promoting personal and professional development; and (v) ensuring a positive environment in which the practice of each social worker can be analyzed and reviewed [29].

Wallbank and Hatton (2011) conclude that supervision in its various forms has been shown to be effective in increasing personal job satisfaction, reducing stress and burnout, and improving the quality of professional services [30,31,32]. Regarding cross-supervision, although work performance is higher when the supervisor is from the same profession, the supervised people perceived trust and a safe environment as more important [33].

Kavanagh, Spence, Strong, Wilson, Sturk, and Crow, identify the key elements of a valuable supervisor as clinical expertise and the ability to provide new and relevant hands-on knowledge and promote learning in a safe and respectful environment [34]. Supervision is by far the most important factor influencing the ethical decision-making process in social work [35].

## 2. Materials and Methods

This paper presents a comparative analysis of the level of burnout in two types of professionals working with people in difficulty in Romania—social workers and clerics. The role of supervision in preventing burnout has also been studied by identifying maximum impact elements in the prevention of burnout. The analysis of burnout was carried out on three distinct underlying dimensions: (i) decreasing personal satisfaction as a result of reducing personal achievements; (ii) emotional burnout; and (iii) depersonalization.

This approach started from a research question regarding the impact of supervision on preventing and controlling burnout in the 7000 social workers and 15,000 clerics working with people in difficulty in Romania.

### Objectives and Hypotheses

The research objectives are:

1. To measure the level of burnout among social workers and clerics in Romania and its correlation with professional supervision.

2. To identify the effects of burnout on professionals working with people in difficulty at the level of personal satisfaction, emotional burnout, and depersonalization.

To achieve these specific objectives, the following research hypotheses have been formulated:

1. Professional working with people in difficulty and not benefitting from quality professional supervision is at higher risk of developing the burnout syndrome. In this study, the level of burnout is determined using the Maslach Burnout Inventory (MBI) [36] in relation to the level of supervision satisfaction determined using the Supervision Quality Assessment Scale (SQAS) [37].

2. Professional working with people in difficulty and with a high score of depressive symptoms is at higher risk of developing the burnout syndrome. High depression score professionals were identified with the Hospital Anxiety and Depression Scale (HADS) [38,39]. The level of burnout was then determined using the MBI.

3. Clerics are at higher risk of developing emotional burnout than social workers. The level of emotional burnout was determined by the corresponding subscale identifying its specific items in the MBI depending on the professional (social worker or cleric).

Burnout is measured using the MBI. This questionnaire was chosen to contrast the Copenhagen Burnout Inventory (CBI) due to its robustness, reliability, and much wider use. At citations level, the ratio is 10:1 in favor of MBI. A score between 25 and 50 shows a low level of burnout; between 51 and 75 shows an average level of burnout; above 75 points shows a high level of burnout.

Although, for a holistic assessment of burnout, a score is taken for the whole questionnaire, there are three dimensions that lead to burnout: emotional burnout consisting of nine items, depersonalization consisting of six items, and the reduction of personal satisfaction as a result of a reduction of professional efficiency and achievements. To eliminate the effects of monotony, eight inverse quotation items were intercalated, i.e., very rarely—5 points, rarely—4 points, sometimes—3 points, frequently—2 points, and very frequently—1 point. Burnout was analyzed both as a whole and per dimensions for a double purpose: on the one hand, to analyze whether the dimensions of burnout correlate negatively with the dimensions of supervision, which is explained in relation to the scale of supervision; and, on the other, to see if, in the case of clerics, a high level of emotional burnout can operate in parallel with elevated levels of personal satisfaction as stated by Barnard and Curry [40].

The second part of the applied questionnaire included the HADS, from which seven items were identified to measure depressive symptoms. This was chosen because it is by far the most used depression measurement scale. Unlike the other depressive-symptom measurement scales containing more than 15 items, it was possible to opt for the HADS as the subscale measures depressive symptoms with seven items, which is useful only as a predictor of burnout, but is not the main purpose of this research.

The third part of the applied instrument included a SQAS containing a scale built from the three functions of a supervisor: administrative, educational, and supportive. Like the MBI, this questionnaire also uses a five-point Likert scale, namely: 1–very rarely, 2–rarely, 3–sometimes, 4–frequently, 5–very frequently. A score between 25 and 50 reflects low quality supervision; between 51 and 75 reflects medium quality supervision; and above 76 points to high quality supervision.

The Q4 scale measures spirituality with a scale (RSS) adapted after Runcan [41], aiming at several measurable aspects of spirituality as effects in relation to God. This scale was chosen to measure more than religiosity and to develop a sufficiently comprehensive scale for both religious cults represented in Romania and professionals who are not religious people.

The research population consisted of 7000 social workers and 15,000 clerics from Romania Romanian-speaking people who have at least one email address. The research sample was non-probabilistic and consisted of a 502-person availability batch, 247 social workers and 255 clerics, with a five-point margin of error. Data collection was accomplished from 13 June to 28 June, 2018, using Google Docs through professional networks. Respondents were professionals from all the counties of Romania. Of course, distribution is not even because of the reluctance or even refusal to supply personal and professional data. The study was conducted according to the guidelines of the Declaration of Helsinki, and approved for publication by the Institutional Review Board of AUREL VLAICU University of Arad, Romania (protocol code 03/3 August 2021). A copy of the questionnaire was provided in Appendix A. The database is freely accessible at http://doi.org/10.3886/E155301V1 (accessed on 23 November 2021).

## 3. Results

The scales applied have a very good internal consistency reliability, measured by Cronbach’s alpha: 0.91 for MBI, 0.75 for HADS, 0.99 for SQAS and 0.89 for RSS. The socio-demographic structure of the sample investigated is shown in Table 1.

The results of the four scales (MBI, HADS, SQAS, and RSS) applied in a comparative cross-tabular between social workers and clerics are shown in Figure 1, Figure 2, Figure 3 and Figure 4 below:

According to these results, there are statistically significant differences between clerics and social workers (chi square = 15.121, df = 2, *p* < 0.001), in the sense that social workers recorded higher burnout scores than clerics (MBI). Additionally, the level of depressive symptoms (HADS) is 9% higher among social workers than clerics (the differences being not statistically significant). Regarding supervision (SQAS), there were statistically significant differences (chi square = 22.999, df = 2, *p* < 0.001) in the sense that clerics recorded significantly higher scores (above 20%) compared to social workers in quality of supervision. A completely different situation between the two sub-samples was obtained in the level of spirituality, the clerics having, as expected, a 40% higher level of spirituality than social workers (chi square = 39.374, df = 2, *p* < 0.001). The comparative analysis through the median and percentiles 25–75 between the scores obtained by the two subgroups (social workers and clerics) in the four scales reflects a clear differentiation (Table 2).

Thus, three of the four scales provide high statistically significant results (*p* < 0.001), i.e., the MBI, the SQAS and the RSS. The fourth scale, the HADS, restricted to depression items, also provides statistically significant results (*p* < 0.05). The results also reflect a higher level of burnout and a higher score of depressive symptoms in social workers than in clerics, while supervision quality and spirituality score higher in clerics than in social workers.

The correlation analysis between these scales reflects a high level of interdependence between them:There is an inversely proportional monotonous relationship between supervision quality and burnout (N = 502, rho = −0.348, *p* < 0.001), which confirms hypothesis no. 1, namely, when supervision is missing or low quality, the risk of burnout increases.There is an inversely proportional monotonous relationship between level of spirituality and level of burnout (rho = −0.343, *p* < 0.001), which mean that when spirituality is missing or of low quality, the risk of emotional burnout increases.There is an inversely proportional monotonous relationship between the administrative function of supervision and the level of emotional burnout (rho = −0.341, *p* < 0.001), which means that when the administrative function of supervision is missing or of low quality, the risk of emotional burnout increases.There is an inversely proportional monotonous relationship between the educational function of supervision and the level of depersonalization (rho = −0.210, *p* < 0.001), which means that when the educational function of supervision is missing or of low quality, the risk of depersonalization increases.There is an inversely proportional monotonous relationship between the supportive function of supervision and the reduction of personal satisfaction (rho = −0.299, *p* < 0.001), which means that when the supportive function of supervision is missing or of low quality, the risk of reducing personal satisfaction increases.

Burnout can be engendered by several factors with different impacts on subjects depending on the labor context and individual features. To quantify the amplitude of the impact of supervision, depressive symptoms, spirituality and socio-demographic variables on burnout, a regression model is used. Thus, the low level of burnout in professionals (social workers and clerics) working with people in difficulty is determined by a high-quality supervision (chi square = 25.361, df = 2, *p* < 0.001), a high level of spirituality (chi square = 5.399, df = 1, *p* < 0.020), and the absence of depressive trends (chi square = 105.596, df = 2, *p* < 0.001). Burnout is not influenced by age or education, but has a significant statistical differentiation for the gender variable (chi square = 10.689, df = 1, *p* < 0.020), women being more prone to burnout than men. By applying a binomial logistic regression model, the distribution shown in Table 3 below is obtained.

In the case of professionals benefiting from high-quality supervision, the probability of not developing burnout is twice as high (OR = 1.962, *p* < 0.05), while in those with a medium and high level of depression, the probability of developing burnout is nine times higher (OR = 8.675, *p* < 0.001). It is also true that quality supervision provided by the supervisor to the professional working with people in difficulty, and assumed by the latter, will protect him/her with a 3 to 1 share of the danger of burnout. The second predictor is of a psychological nature, namely depressive symptoms. A medium or high score indicates the risk of burnout syndrome with a 5–17 to 1 share. Among personal factors, the spiritual factor has a positive influence in preventing personal burnout syndrome, along with individual factors, such as increased levels of education, age, or seniority. Women also run a higher risk of burnout syndrome than men.

## 4. Discussion

In this research, the goal was to analyze the role of supervision in two categories of professionals in Romania working with people in difficulty, namely social workers and clerics to prevent burnout. Analyzing burnout in two categories of professionals, there is a statistically significant difference between clerics and social workers, in the sense that social workers had higher burnout scores than clerics.

Measuring the level of depressive symptoms, it was clear that only three of the seven items had statistically significant outcomes, while three other items only approached the significance threshold. Expressed as a percentage, the level of depressive symptoms among social workers is 9% higher than in clerics. As regards supervision, the results were statistically significant, in the sense that clerics recorded 20% higher scores in the quality of supervision than social workers. The results were also statistically significant in terms of spirituality: as expected, clerics had 40% higher scores than social workers.

Comparisons between groups (made with the Mann-Whitney U test) show a higher level of burnout and a higher score of depressive symptoms in social workers than in clerics. Three of the four scales produced strong statistically significant outcomes (*p* < 0.001), i.e., Scale 1 (MBI), Scale Q3 (SQAS) and Scale 4 (RSS), while Scale 2 (HADS), restricted to depression items, also provides statistically significant results (*p* < 0.05).

Correlation analyses have also shown that there is an inversely proportional medium power relationship (rho = −0.348) between the level of burnout and supervision quality (*p* < 0.001). When supervision is missing or low quality, the risk of burnout increases; and vice versa, when supervision is high quality, the risk of burnout decreases. There is also an inversely proportional medium power relationship (rho = −0.343) between the level of burnout and the level of spirituality (*p* < 0.001). When spirituality is missing or low quality, the risk of burnout increases; and vice versa, when spirituality is high quality, the risk of burnout decreases.

As for the functions of supervision, there is an inversely proportional medium power relationship (rho = −0.341) between the level of emotional burnout and the administrative function of supervision (*p* < 0.001); that is, when the administrative function of supervision is missing or low quality, the risk of emotional burnout increases. There is also an inversely proportional low-quality power relationship (rho = −0.210) between the level of depersonalization and the educational function of supervision (*p* < 0.001); that is, when the educational function of supervision is missing or low quality, the risk of depersonalization increases. On the other hand, there is an inversely proportional monotonous power relationship (rho = −0.299) between the level of reduction of personal satisfaction and the supportive function of supervision (*p* < 0.001), i.e., when the supportive function of supervision is missing or low quality, personal satisfaction decreases.

There is also a higher level of emotional burnout among social workers than clerics, the differences between the clerics’ score (2.00) and the social workers’ score (2.22) being statistically significant (*p* < 0.001). Following the regression analysis, it can also be concluded that professionals working with people in difficulty who have a high score of depressive symptoms run a 5–17 times greater risk of developing a burnout syndrome than professionals who do not show depressive symptoms. Complementarily, quality supervision provided and assumed by professionals working with people in difficulty will provide three times the amount of protection from the risk of burnout. Among personal factors, the spirituality factor has a positive influence in preventing the burnout syndrome, along with education, age, or seniority. Women also run a higher risk of burnout syndrome than men.

One of the main limitations of this research is that the conclusion represent mostly general trends and the data cannot be generalized, as the sample was a non-probabilistic one. Also, the office workers, not being immediately considered as caregivers, might have lower values in some burnout sub-scales as taken from the MBI.

As other limits of the study, it should be noted that there are variables that can influence burnout in professionals working with people in difficulty, which were not included in this study but that could be the subject of future research. Regarding supervision, interviewing or mentoring, although the satisfaction of the professional supervised was measured and compared, there may be some differences between the supervision or interviewing of the social workers and the mentoring of clerics. Even in certain clerical traditions, mentoring may have certain features influencing its effectiveness just like, in the supervision of different categories of social workers, there are various schools with different focuses. The history of supervision in supportive professions shows that each profession develops its own tradition and literature in the field [42,43,44].

## 5. Conclusions

Finally, it can be said that burnout in social workers and clerics in Romania working with people in difficulty can be prevented primarily through professional supervision, a profession that is, unfortunately, at an incipient stage of development in Romania. At present, in the two professions, the function of supervision is taken over by interviewing (by colleagues) and mentoring (especially in clerics, by a colleague, a hierarchical superior, or even a confessor). The conclusions of this research confirm the results of other studies on professionals who work with people in difficulty in other countries and cultures, as well as the need for quality supervisions in three areas: administrative, educational, and supportive to prevent burnout.

A longitudinal study of supervision and spirituality influence would also provide valuable information on the prevention of burnout in the professionals working with people in difficulty, namely social workers and clerics. One of the necessary research directions in the future is, according to Carpenter [3] the feedback from beneficiaries of social services regarding the effects of the professional supervision of social services. The conclusions in this area come from correlations made mainly in the U.S.A. on the positive results from these services (idem).

## Figures and Tables

**Figure 1 ijerph-19-00160-f001:**
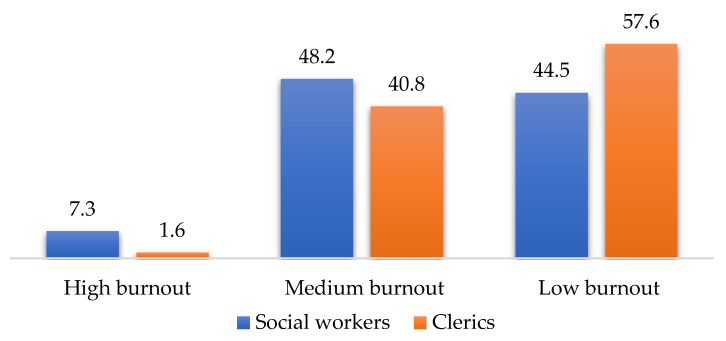
MBI scores for social workers and clerics.

**Figure 2 ijerph-19-00160-f002:**
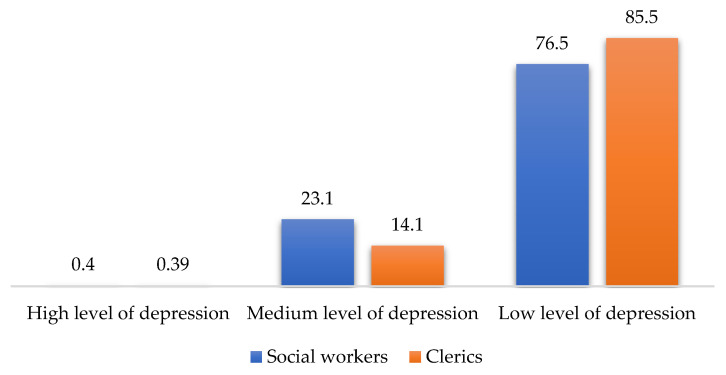
HADS scores for social workers and clerics.

**Figure 3 ijerph-19-00160-f003:**
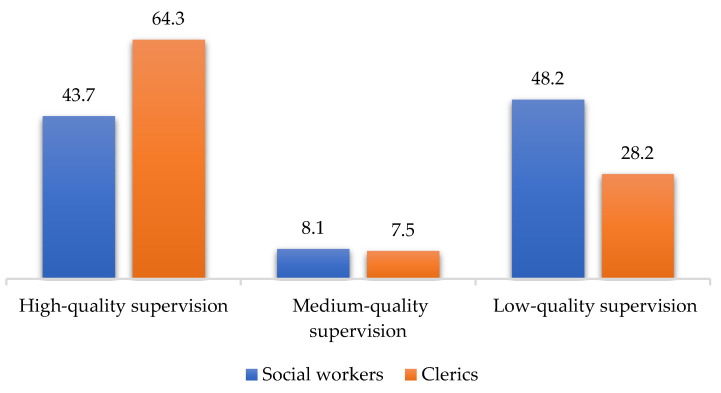
SQAS scores for social workers and clerics.

**Figure 4 ijerph-19-00160-f004:**
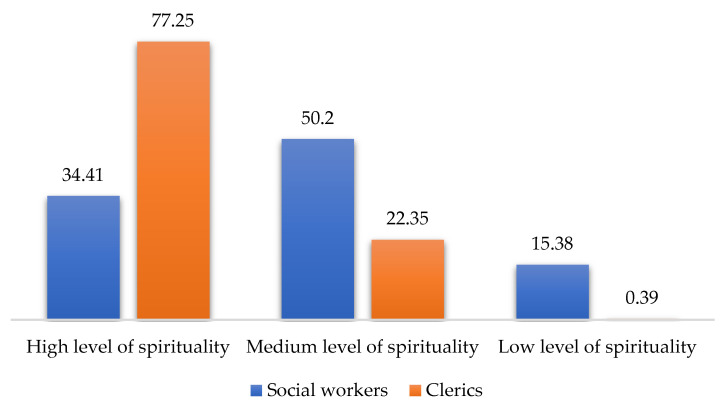
RSS scores for social workers and clerics.

**Table 1 ijerph-19-00160-t001:** Socio-demographic structure of the sample.

Factual Data		Respondents	
		N	%
Profession	social workers	247	49.2
	clerics	255	50.8
Gender	male	229	45.6
	female	273	54.4
Age	21–30 years	66	13.1
	31–40 years	168	33.5
	41–50 years	182	36.3
	51–60 years	76	15.1
	60+ years	10	2.0
Education	high school	34	6.8
	college	208	41.4
	master’s degree	234	46.6
	doctor’s degree	26	5.2
Seniority	0–5 years	105	20.9
	6–15 years	166	33.1
	16–30 years	204	40.6
	30+ years	27	5.4
Work environment	rural	91	18.1
	urban	411	81.9
Marital status	married	430	85.7
	single	72	14.3

**Table 2 ijerph-19-00160-t002:** Comparative analysis of the median and percentiles 25–75 between the scores obtained in the four scales by the two subgroups (social workers and clerics).

Clergy/Social Worker			Percentiles						
			5	10	25	50	75	90	95
Weighted Average (Definition)	MBI score	C	30.80	33.00	40.00	48.00	57.00	64.00	70.00
		SW	33.00	35.80	42.00	54.00	63.00	71.20	78.60
	HADS score	C	0.0000	0.1429	0.2857	0.4286	0.7143	1.0000	1.2857
		SW	0.0000	0.1429	0.2857	0.5714	0.8571	1.2857	1.5143
	SQAS score	C	0.00	0.00	0.00	94.00	107.00	120.00	125.00
		SW	0.00	0.00	0.00	61.00	99.00	115.20	123.00
	RSS score	C	16.60	18.00	21.00	24.00	26.00	28.00	29.00
		SW	5.40	8.00	12.00	17.00	22.00	26.00	28.00
Test Statistics ^a^		MBI Score		HADS Score		SQAS Score		RSS Score	
Mann-Whitney U		24,948.500		28,263.000		23,822.500		13,809.000	
Wilcoxon W		57,588.500		60,903.000		54,450.500		44,437.000	
Z		−4.029		−2.001		−4.826		−10.900	
Asymp. Sig. (2-tailed)		0.000		0.045		0.000		0.000	

^a^ Grouping Variable: C/AS.

**Table 3 ijerph-19-00160-t003:** Factors influencing burnout in social workers and clerics.

	B	*p*	OR	95% CI for OR	
				Min	Max
Levels of supervison quality		0.004			
medium-quality supervision	−0.161	0.700	0.851	0.375	1.933
high-quality supervision	0.674	0.003	1.962	1.264	3.045
Medium and high depression	2.160	0.000	8.675	5.334	14.108
Gender (1)	0.395	0.271	1.484	0.735	2.997
Age (three intervals)		0.987			
Age interval (1)	0.024	0.917	1.024	0.656	1.598
Age interval (2)	0.045	0.881	1.046	0.579	1.891
Education		0.268			
Education (1)	0.183	0.675	1.200	0.511	2.817
Education (2)	0.486	0.263	1.626	0.695	3.805
Level of spirituality (1)	0.449	0.272	1.566	0.703	3.487
Social workers and clerics	0.073	0.839	1.076	0.531	2.180
Constant	−0.920	0.000	0.398		

## Data Availability

The database is freely accessible at http://doi.org/10.3886/E155301V1 (accessed on 23 November 2021).

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
