# Peer review of "The Role of Supervision in Preventing Burnout among Professionals Working with People in Difficulty"

_ijerph, 2021, doi:10.3390/ijerph19010160_

Round 1
Reviewer 1 Report
I have read through the paper with interest; while interesting, I believe this manuscript should be intensively revised.
The introduction is far too long and should be trimmed to 2-3 tight paragraphs that get to the point. I also disagree with the position of the 3 research objectives and the 7 research hypotheses (instead of the materials and methods section, this reviewer suggests ending the introduction section with the 1 hypothesis and 1 -or 2- objective/s).
Also, the limits of the study paragraph should be in the discussion section, rather than at conclusions.
Although the approach started with 7,000 social workers and 15,000 clerics working with people in difficulty in Romania, the research sample consisted of 502 subjects. Could you include the sample size calculation?
Also, it is stated that the database is free to access, but could you provide the reference or refer to OPEN ICPSR?
The Institutional Review Board Statement was approved on 03.08.2021? And all the web pages of the reference section accessed on 11.18.2021? Please, clarify. At the references, some of these should be reviewed and standardized. E.g., reference 8 includes forenames and surnames without order. This reviewer would also suggest not to include references 1 and 29 (dictionary.com and yourdictionary.com, respectively).
Author Response
Thank you very much for your comments and observations, there are really helpful for improving our paper. Below, are our answers to each point:
I have read through the paper with interest; while interesting, I believe this manuscript should be intensively revised.
The introduction is far too long and should be trimmed to 2-3 tight paragraphs that get to the point.
We have tried very hard to have an as short as possible introduction (theoretical model), and we have made it 2 pages… If the editor also recommends to reduce it to 2-3 paragraphs, we will rewrite it, but our fear is that it will lack the connections with relative concepts and, also, the national context of the phenomenon. We really need a very clear approach of the what we understand by “professional working with people in difficulty”, the particularity of the national system, and the role of supervising.
I also disagree with the position of the 3 research objectives and the 7 research hypotheses (instead of the materials and methods section, this reviewer suggests ending the introduction section with the 1 hypothesis and 1 -or 2- objective/s).
We have reduced the objectives to 2 and the hypotheses to 3.
Also, the limits of the study paragraph should be in the discussion section, rather than at conclusions.
We have moved the paragraph referring to the Limits to the Discussion.
Although the approach started with 7,000 social workers and 15,000 clerics working with people in difficulty in Romania, the research sample consisted of 502 subjects. Could you include the sample size calculation?
We have detailed that it was a convenience sample and not a probabilistic one.
Also, it is stated that the database is free to access, but could you provide the reference or refer to OPEN ICPSR?
It is included in lines 251-252
The Institutional Review Board Statement was approved on 03.08.2021?
The Institutional Review Board Statement was for publishing the data (we have presented the entire methodology and the type of results).
And all the web pages of the reference section accessed on 11.18.2021? Please, clarify.
It was a computer mistake, we have adjusted.
At the references, some of these should be reviewed and standardized. E.g., reference 8 includes forenames and surnames without order.
We have made all the corrections.
This reviewer would also suggest not to include references 1 and 29 (dictionary.com and yourdictionary.com, respectively).
We have made these corrections, too.
Reviewer 2 Report
The article deals with a research conducted on 502 subjects in the Romanian context in which the role of supervision on Burnout is explored, specifically differentiating the situation between social workers and clericals. Although the work is interesting and correctly structured, I have doubts about some aspects that need to be clarified before I can proceed with the next steps. In particular:
1) The biggest limitation of the study, in my opinion, is that it lacks a reference theory to justify the assumptions of the hypotheses. Why should supervision reduce Burnout risk? I suggest introducing a theoretical model that could explain this effect;
2) Line 23 of the abstract: What does sociological questionnaire mean? It would be better to define it as either quantitative or qualitative;
3) Check that the citations are in the correct format: for example in line 43 [4] is before the full stop.
4) Line 51: COVOD or COVID?
5) Line 109-128: is it necessary to insert 18 lines of inverted commas?
6) Line 147: I would insert a sub-section called "Objectives and hypotheses";
7) A section on the socio-demographic characteristics of the sample is completely missing. Gender, educational qualification, average age? This is all essential information to be able to contextualise the results obtained;
8) How is it possible to define the sample as "representative" if you state in line 222 that it is non-probabilistic?
9) Line 233: It is preferable to use only two decimal places;
10) Table 3: Levels, not levers;
11) In the conclusions and in particular in the limitations, besides the obvious considerations about the impossibility of generalising the results as the sample is NOT probabilistic, it should also be pointed out that office workers, not being immediately considered as caregivers, might have lower values in some Burnout sub-scales as taken from the MBI. This is an important limitation of the study that must be taken into account.
12) I suggest an important English proofreading of the paper.
Author Response
Thank you very much for your comments and observations, there are really helpful for improving our paper. Below, are our answers to each point:
The article deals with a research conducted on 502 subjects in the Romanian context in which the role of supervision on Burnout is explored, specifically differentiating the situation between social workers and clericals. Although the work is interesting and correctly structured, I have doubts about some aspects that need to be clarified before I can proceed with the next steps. In particular:
1) The biggest limitation of the study, in my opinion, is that it lacks a reference theory to justify the assumptions of the hypotheses. Why should supervision reduce Burnout risk? I suggest introducing a theoretical model that could explain this effect;
Due to the limitation of the article structure (the other reviewer has made the observation that the introduction is too long), this aspect is presented as concisely as possible. We have added 3 more paragraphs, because indeed, it is the core theoretical point of our study.
2) Line 23 of the abstract: What does sociological questionnaire mean? It would be better to define it as either quantitative or qualitative;
By default, a sociological questionnaire is a research tool for a quantitative approach (survey), and it is included in Appendix A (411-469). The interview guide is the equivalent tool (similar with questionnaire) for a qualitative approach. If you consider it necessary, we can develop this idea in the abstract
3) Check that the citations are in the correct format: for example in line 43 [4] is before the full stop.
We have made all the corrections.
4) Line 51: COVOD or COVID?
Done…
5) Line 109-128: is it necessary to insert 18 lines of inverted commas?
It is an extensive definition of “people in difficulty”, we made a little adjustment.
6) Line 147: I would insert a sub-section called "Objectives and hypotheses";
Done
7) A section on the socio-demographic characteristics of the sample is completely missing. Gender, educational qualification, average age? This is all essential information to be able to contextualise the results obtained;
The information is in Table 1
8) How is it possible to define the sample as "representative" if you state in line 222 that it is non-probabilistic?
Indeed, it was a non-probabilistic sampling with a convenience sample, we make the correction.
9) Line 233: It is preferable to use only two decimal places;
We have made the changes.
10) Table 3: Levels, not levers;
Done
11) In the conclusions and in particular in the limitations, besides the obvious considerations about the impossibility of generalising the results as the sample is NOT probabilistic, it should also be pointed out that office workers, not being immediately considered as caregivers, might have lower values in some Burnout sub-scales as taken from the MBI. This is an important limitation of the study that must be taken into account.
We have included these observations too. (392-395)
12) I suggest an important English proofreading of the paper.
We have asked for help from a professional service from UK.
Round 2
Reviewer 1 Report
Author's reply include almost all the topics addressed for revision, so this reviewer's recommendation is to accept the paper in the present form